# The Pandemic Allocation of Ventilators Model Penalizes Infants with Bronchopulmonary Dysplasia

**DOI:** 10.3390/children10081404

**Published:** 2023-08-17

**Authors:** Anupama Sundaram, Jonathan M. Fanaroff, Deanne Wilson-Costello, Melissa Alberts, Naini Shiswawala, Noam Stern, Rita M. Ryan

**Affiliations:** Department of Neonatology, University Hospitals Rainbow Babies and Children’s Hospital, Case Western Reserve University, Cleveland, OH 44106, USA

**Keywords:** bronchopulmonary dysplasia, triage, resource allocation

## Abstract

During the COVID-19 pandemic, institutions developed ventilator allocation models. In one proposed model, neonates compete with adults for ventilators using a scoring system. Points are given for conditions that increase one- and five-year (y) mortality. For example, comparable points were added for adult conditions with mortality of 71.3% and for neonates with moderate or severe bronchopulmonary dysplasia (mod/sBPD). We hypothesized that this model overestimates mortality in neonates with BPD and would penalize these infants unfairly. There was little information available on 1 y and 5 y mortality risk for mod/sBPD. To evaluate this allocation protocol, a retrospective chart review was performed on infants born ≥22 weeks and weighing <1500 g admitted to Rainbow Babies and Children’s Hospital in 2015 to identify babies with BPD. The main outcomes were 1 and 5 y mortality. In 2015, 28 infants were diagnosed with mod/s BPD based on NIH 2001 definition; 4 infants had modBPD and 24 had sBPD. All infants (100%) with modBPD survived to 5 y; 2 infants with sBPD died by 1 y (8%) and 22 survived (92%) to 1 y; 3 died (12.5%) by 5 y; and at least 13 survived (54%) to 5 y. Infants with mod/s BPD had lower-than-predicted 1 and 5 y mortality, suggesting the points assigned in the model are too high for these conditions. We believe this model would unfairly penalize these babies.

## 1. Introduction

In response to the COVID-19 pandemic, to appropriately distribute limited resources including ventilators, if necessary, institutions developed ventilator allocation models. During this time, health care goals shifted from the individual patient to benefit the greatest number of individuals. Allocation models were developed to facilitate fair and evidence-based decision making. The policies developed applied to all patients in need of critical care resources. This created a situation in which adults would compete with neonates for ventilators. The two main approaches to triage include maximizing lives and maximizing life years. Choosing between these two approaches has implications for neonatology. Some adults who need ventilators will have a survival rate higher than some extremely premature babies. However, surviving babies will likely live longer, maximizing life years [1]. Certain states created guidance to specifically avoid the maximization of life years approach. Their concerns included discrimination based on age since projections on maximization of “life- years is difficult to predict and could be used against treatment of older adults” [2]. The life years approach was not applied in our regional protocols. A limited number of states did create recommendations for the allocation of resources; 26 states have publicly available guidelines. However, there was often a lack of guidance on pediatric or neonatal allocation. A systematic review concluded that only 14 states had pediatric guidelines included in their recommendations [3]. It was noted that there was significant variation in allocation among these states. The variation in pediatric and neonatal triage guidelines is concerning and creates inequity in resource allocation across states. Also, neonates with respiratory failure would only be competing with adults for a conventional ventilator. Other forms of invasive ventilation, such as high-frequency oscillation or a jet ventilator, could be used in neonates instead of a conventional ventilator, but are not routinely used in adults.

In our hospital’s proposed model, which was planned but never implemented, neonates compete with adults for ventilators using a scoring system based on predicting survival. In this allocation model based on the Yale New Haven Health System triage protocol [4], neonatal ventilators are given to adults with acute respiratory failure from COVID-19, for example, if they have a triage score indicating a higher chance of survival than a neonate. In our hospital’s model, triage scores incorporate severity of illness scoring based on the specific population: short term, i.e., one-year mortality, and long term, i.e., five-year mortality (Figure 1). Triage scores are assigned from one to eight, with one being the highest priority for ventilator allocation. During a public health emergency, patients must have a triage score of less than six to be eligible for critical care support.

Adults are assigned scores based on their Sequential Organ Failure Assessment (SOFA) score [5] and associated co-morbid conditions; children are assigned scores using the Pediatric Logistic Organ Dysfunction-2 score (PELOD-2) and associated co-morbid conditions [6]. Neonates are assigned scores based on either the National Institute of Health (NIH) Extremely Preterm Birth Outcomes Tool [7] or the Score for Neonatal Acute Physiology with Perinatal Extension (SNAPPE-II) [8], along with associated co-morbid conditions. Both of the neonatal scoring tools are not validated for use in a triage setting. They were designed to assess outcomes in the setting of periviability and applied in the first 24 h after birth. In our institution’s proposed allocation model, additional points are given to co-morbid conditions that increase one- and five-year mortality. Two points are added for increased five-year mortality and four points are given for increased one-year mortality. To avoid discriminating against individuals based on age, disability, or pre-existing disease, the model was designed so that the only outcome of importance for the scoring tools and additional points was mortality.

As survival of preterm infants has increased over the last three decades, there has been a notable increase in the rates of bronchopulmonary dysplasia (BPD) in babies born at a 22–29 weeks’ (w) gestational age from 32% in 1993 to 40% by 2008 [9]. The incidence of BPD continues to increase with a reported 45.5% of surviving preterm infants being diagnosed with BPD [10]. With BPD being a major cause of morbidity and mortality in surviving preterm infants, this condition was included (among others) as a neonatal co-morbid condition in the triage protocol. As per the NIH 2001 definition [11], BPD is defined on a scale of mild, moderate, and severe based on the need for supplemental oxygen and degree of respiratory support at a postmenstrual age of 36 w. Moderate BPD is defined as 22–30% oxygen at a postmenstrual age of 36 w. Severe BPD is >30% oxygen, or on continuous positive airway pressure (CPAP), or on a mechanical ventilator at a postmenstrual age of 36 w. Infants who have a diagnosis of moderate bronchopulmonary dysplasia (BPD) are assigned two points for increased five-year mortality. Infants who have severe bronchopulmonary dysplasia are assigned four points for increased one-year mortality.

An example of an adult condition receiving four points for increased one-year mortality in the model is advanced cirrhosis (Table 1). When we investigated the literature, this condition was associated with a one-year mortality of 71.3% [12]. Our experience would not have assigned a mortality rate this high to babies with either moderate BPD or severe BPD. An example of an adult condition receiving two points for increased five-year mortality in the model is malignancy with a five-year mortality of 100%. This mortality rate comparison also seemed unfair for babies with moderate BPD. When we began our study, we could not find reliable data concerning the one-year or five-year mortality rates for babies with BPD, so our goal was to determine these rates from our own population. We hypothesized that this planned allocation model overestimates mortality in neonates with moderate or severe BPD and would inappropriately penalize these infants.

## 2. Materials and Methods

To evaluate whether this current model for triage allocation correctly predicts long-term mortality in infants with moderate-to-severe BPD, a single-center retrospective chart review was performed. All infants born ≥22 weeks and weighing <1500 g who were admitted to Rainbow Babies and Children’s Hospital in 2015 were screened to identify babies with moderate or severe BPD. This retrospective study was approved by the University Hospitals (UH) Institutional Review Board with consent waived. The main outcomes were one- and five-year mortality. Data collection was performed using the electronic medical record by physicians who collected demographic and clinical data. By design, follow-up was ascertained only for infants with later visits of any type to the broader University Hospital’s system. A sample size calculation was based on 1 y survival and indicated a need of at least 15 patients with known mortality outcomes to show a significant difference of 30% between BPD-related mortality and adult co-morbid condition mortality to reject the null hypothesis of no difference between BPD-related mortality compared to adult co-morbid condition mortality with a power of 80% and an alpha value of 0.05.

## 3. Results

In 2015, twenty-eight infants were diagnosed with moderate-to-severe BPD at Rainbow Babies and Children’s Hospital (RBC) Neonatal Intensive Care Unit (NICU). Of those twenty-eight infants, four infants had moderate BPD and twenty-four infants had severe BPD (Figure 2). An assessment of one-year and five-year mortality in babies with moderate BPD showed that all infants (100%) with moderate BPD were alive at one year and at five years of age (Table 1). An assessment of one-year mortality in infants with severe BPD showed that, by one year, two had died (8%) and twenty two survived (92%). We also included an assessment of five-year outcome for babies with severe BPD, even though the triage protocol only utilized one-year mortality rates for these babies. For these twenty-four babies with severe BPD, by five years, three infants had died (12.5%) and at least thirteen survived (54%). We did not have five-year outcome for eight infants with severe BPD (33%). Thus, the lowest possible survival was 54% for babies with severe BPD at five years, if we assume that the babies without follow-up data did not survive.

## 4. Discussion

Infants with moderate-to-severe BPD had lower-than-predicted (through the allocation model) one- and five-year mortality, suggesting the points assigned in the model are too high for these conditions. Triage protocols created during the pandemic aimed to distribute limited resources to the patients with the best chance of survival. However, in the proposed model there were significant mortality differences between the two populations assigned the same co-morbidity points when we examined the mortality outcomes for babies with BPD. Infants with moderate BPD were assigned two points for increased five-year mortality, but we had 100% survival/0% mortality. This was a low n of only four babies, but in two papers we found after we completed the study, the authors reported 52% mortality at most (they also had incomplete ascertainment) for moderate-BPD babies at five years, which is much lower than the reported mortality for adult conditions with comparable points (Table 1) [14]. Infants with severe BPD were assigned four points for increased one-year mortality. However, our population showed lower mortality rates at both one and five years. The current triage model underestimates the survival of these neonatal co-morbid conditions.

Our findings show that more investigation is needed in the area of triage for neonates. The survival of preterm infants has increased over the last three decades due to improvements in neonatal intensive care [9]. Infants who survive the first days of life have been shown to have a progressively greater likelihood of survival with each passing NICU day [15]. As death and BPD are competing outcomes, with the improvement in survival, there was a notable increase in rates of BPD. With BPD being a major cause of morbidity and mortality in surviving preterm infants, it was incorporated into our hospital’s triage protocol. Rather than a longer term such as one-year or five-year survival, most prior studies assessing BPD mortality evaluate and report survival to discharge home. This is a typical parameter for evaluating mortality in preterm infants. A study evaluating mortality in BPD subjects born from 1995 to 2008 showed an overall mortality rate of 12.8% prior to hospital discharge [16]. This outcome measure is different than the one- and five-year survival outcome measures used during the development of our hospital system’s allocation protocol. We found limited studies evaluating BPD mortality at five years. Thus, if co-morbid conditions are to be assessed in neonates when assembling a triage protocol, local mortality data could be important to fairly distribute limited resources. In continuing to monitor the literature in this area, after our data collection, we found newly published data evaluating long-term BPD mortality in outpatients with bronchopulmonary dysplasia noting an overall mortality rate of <1% for the first year of age, then an annual mortality rate of 0.037% for ages 1–4 [17]. In our patient population, with our severe BPD 1-year mortality at 8% (receiving the same points as the adult condition with 71% mortality) and moderate BPD 5-year mortality at 0% (receiving the same mortality as the adult condition with 100% mortality), we believe the proposed model would have unfairly penalized premature babies with BPD.

Santini and colleagues discussed other possible strategies for utilizing other ventilators (e.g., anesthesia equipment while operating rooms were closed) and decreasing the need for ventilators in adults [18]. We noted as this policy was being developed that the neonatal population has some advantages. Babies on high-frequency ventilators (HFV) (jet or oscillator) would not be at risk for competing with other populations for a conventional ventilator, nor would a baby on equipment to provide non-invasive ventilation that is unique to the neonatal population, such as some of the biphasic and other non-invasive ventilation devices. Our first preference would be to have babies on these other ventilators or devices as appropriate to free up conventional ventilators for older children and adults. We used HFV generally as a rescue mode but babies could be moved from CV to HFV easily if the need arose. During an emergency that requires actually allocating ventilators, with neonates competing with other populations, we would have to be as creative as possible.

Treatment strategies have been evolving over decades for the management of bronchopulmonary dysplasia, especially with changes in treatment for respiratory distress syndrome (RDS). Early continuous positive airway pressure (CPAP) has reduced the need for mechanical ventilation in preterm infants with respiratory distress syndrome compared to the previous practice of intubation and mechanical ventilation in the delivery room. However, with these changes in practice, the rates of bronchopulmonary dysplasia have not decreased [9]. Nasal intermittent positive pressure ventilation (NIPPV) is being used in place of CPAP as an improved method for keeping infants on non-invasive respiratory support following extubation, with studies currently concluding no difference seen in the rates of bronchopulmonary dysplasia between the groups [19]. As we continue to learn and understand this disease process, future targeted therapy may decrease the need for mechanical ventilation in this population.

The limitations of this study include small sample size, single-center design, and incomplete outcome ascertainment. Further investigation is ongoing to address these limitations and to apply the base model score to our NICU patients “in real time”. We hope to use our findings to inform future pandemic triage protocols. We believe that local mortality rates should be used when available for a ventilator allocation protocol.

## 5. Conclusions

Thankfully, the COVID-19 pandemic did not require the implementation of the ventilator allocation protocol at our institution. The COVID-19 pandemic has demonstrated the importance of triage planning in advance. Our results from the retrospective portion of our study demonstrate that this model would have unfairly disadvantaged NICU patients. It is important to identify and develop screening measures which are useful for triage in neonates so that an established tool is available for providers and medical systems to design fair protocols to allocate critical resources.

## Figures and Tables

**Figure 1 children-10-01404-f001:**
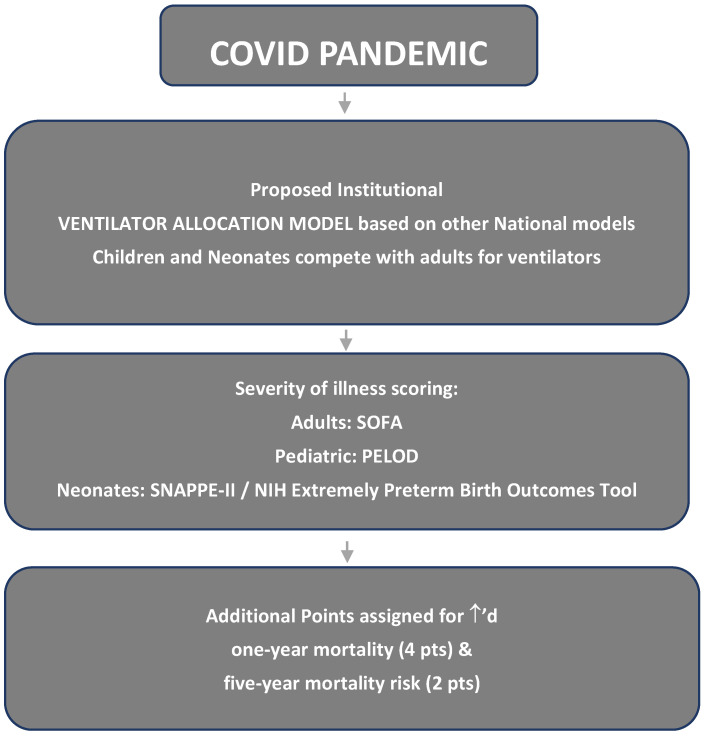
Ventilator allocation guideline.

**Figure 2 children-10-01404-f002:**
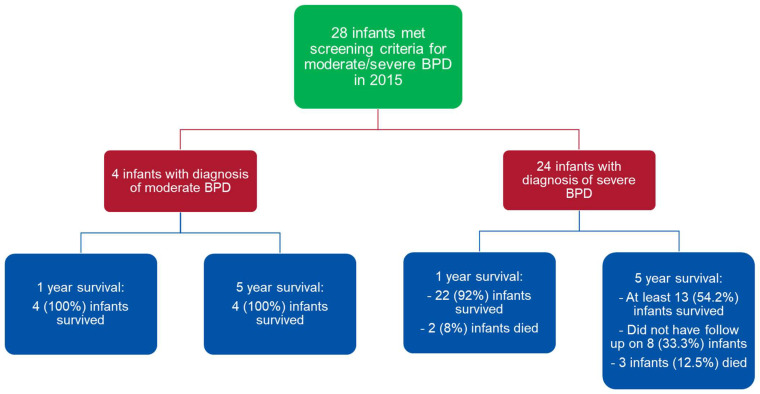
Outcome of subjects according to disease severity.

**Table 1 children-10-01404-t001:** Comparing the ventilator allocation protocol for the COVID-19 pandemic.

	1-Year Mortality	Points Assigned for Death Likely in 1 Year	5-Year Mortality	Points Assigned for Substantial Impact on Survival
Adult condition: Cirrhosis with MELD > 20	Maximum reported 71.3% [12]	4	NA	NA
Adult condition Malignancy with less than 5 year survival	NA	NA	100%	2
BPD mortality based on the literature Moderate BPD Severe BPD	NA	NA	At most 52% * [13]	2
	15.8% [14]	4	NA	NA
Our population Babies <1500 g born in 2015 Moderate BPD Severe BPD	0%	NA	0%	2
	8%	4	12.5%-at most 45.8%	NA

BPD, bronchopulmonary dysplasia: moderate <30% supplemental oxygen at a postmenstrual age of 36 w, severe BPD <30% supplemental oxygen, on CPAP, or ventilator at a postmenstrual age of 36 weeks. MELD: Model for end-stage liver disease. NA: Not applicable. * Similar to our study, some infant follow-up data were lost.

## Data Availability

The datasets generated during and/or analyzed during the current study are not publicly available due to HIPPA, but a de-identified dataset is available from the corresponding author on reasonable request.

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
