# Peer review of "The Pandemic Allocation of Ventilators Model Penalizes Infants with Bronchopulmonary Dysplasia"

_children, 2023, doi:10.3390/children10081404_

Round 1
Reviewer 1 Report
Dear Authors,
It was my pleasure reviewing your well written and insightful research study. However, I do have few questions and suggestions;
1. Could you explain why you chose the year 2015 for the study?
2. The sample size of 28 patients in 2015 seem too small. don't you think extending the study period from 2015 to 2019 will increase the sample size and yield a strong significant result? if not, why not?
Author Response
Thank you for reviewing our paper. We choose 2015 as the study year because we started the study in 2020 and need five year mortality outcome. We did not want to go back further to add more babies because we felt practices would have changed even more the further we went back.
Reviewer 2 Report
Dear colleagues, thank you for the interesting paper.
In page 3 you are stating "As survival of preterm infants has increased over the last three decades, there has been a notable increase in the rates of bronchopulmonary dysplasia (BPD) in babies born 22-29 weeks’ (w) gestational age from 32% in 1993 to 40% by 2008 [9]".
There are more recent data documenting an increase of BPD likelihood, in very small premature babies, in spite of new, more gentle ventilation modes and tailored approach.
Yung et colleagues [Jung YH, Jang J, Kim HS, Shin SH, Choi CW, Kim EK, Kim BI. Respiratory severity score as a predictive factor for severe bronchopulmonary dysplasia or death in extremely preterm infants. BMC Pediatr. 2019 Apr 23;19(1):121. doi: 10.1186/s12887-019-1492-9. Erratum in: BMC Pediatr. 2019 Jul 31;19(1):263. PMID: 31014304; PMCID: PMC6480897.] are presenting in a 2019 paper data from a cohort 2010-2014 with 41.3% rate of severe BPD (57 out of 138 patients).
Xueyu Chen et colleagues [Chen X, Li H, Qiu X, Yang C, Walther FJ. Neonatal hematological parameters and the risk of moderate-severe bronchopulmonary dysplasia in extremely premature infants. BMC Pediatr. 2019 Apr 30;19(1):138. doi: 10.1186/s12887-019-1515-6. PMID: 31039810; PMCID: PMC6489335.] are producing a 50% of BPD in extremely preterm infants.
Jensen and coworkers are stating in a 2021 Pediatrics paper [Jensen EA, Edwards EM, Greenberg LT, Soll RF, Ehret DEY, Horbar JD. Severity of Bronchopulmonary Dysplasia Among Very Preterm Infants in the United States. Pediatrics. 2021 Jul;148(1):e2020030007. doi: 10.1542/peds.2020-030007. Epub 2021 Jun 2. PMID: 34078747; PMCID: PMC8290972.] that in a cohort of 24.896 US premature born-infants 22-29 weeks gestational age "More than one-half of very preterm infants born in the United States died before 36 weeks' PMA or developed BPD."
In page 6 you are stating "Infants who survive the first days of life have been shown to have a progressively greater likelihood of survival with each passing NICU day [12]". There are several attempts to generate predictive tools to anticipate outcomes and several biomarkers were proposed for this cohort of children in order to document risk.
Such tools are:
1. airway microbiome [Gentle SJ, Lal CV. Predicting BPD: Lessons Learned From the Airway Microbiome of Preterm Infants. Front Pediatr. 2020 Feb 4;7:564. doi: 10.3389/fped.2019.00564. PMID: 32117822; PMCID: PMC7011099.],
2. umbilical cord blood vitamin D levels [Yu H, Fu J, Feng Y. Utility of umbilical cord blood 25-hydroxyvitamin D levels for predicting bronchopulmonary dysplasia in preterm infants with very low and extremely low birth weight. Front Pediatr. 2022 Aug 4;10:956952. doi: 10.3389/fped.2022.956952. PMID: 35989993; PMCID: PMC9386287. or Lu T, Liang B, Jia Y, Cai J, Wang D, Liu M, He B, Wang Q. Relationship between bronchopulmonary dysplasia, long-term lung function, and vitamin D level at birth in preterm infants. Transl Pediatr. 2021 Nov;10(11):3075-3081. doi: 10.21037/tp-21-494. PMID: 34976773; PMCID: PMC8649600.]
3. non-invasive sampling approaches to monitoring [Cui X, Fu J. Early prediction of bronchopulmonary dysplasia: can noninvasive monitoring methods be essential? ERJ Open Res. 2023 Apr 3;9(2):00621-2022. doi: 10.1183/23120541.00621-2022. PMID: 37020839; PMCID: PMC10068511.]
4. genetic background of child [Hamvas A, Feng R, Bi Y, Wang F, Bhattacharya S, Mereness J, Kaushal M, Cotten CM, Ballard PL, Mariani TJ; PROP Investigators. Exome sequencing identifies gene variants and networks associated with extreme respiratory outcomes following preterm birth. BMC Genet. 2018 Oct 20;19(1):94. doi: 10.1186/s12863-018-0679-7. PMID: 30342483; PMCID: PMC6195962.]
Risk scoring tools validated for all ethnicities and for all preterm infants were recently published [Yu Z, Wang L, Wang Y, Zhang M, Xu Y, Liu A. Development and Validation of a Risk Scoring Tool for Bronchopulmonary Dysplasia in Preterm Infants Based on a Systematic Review and Meta-Analysis. Healthcare (Basel). 2023 Mar 6;11(5):778. doi: 10.3390/healthcare11050778. PMID: 36900783; PMCID: PMC10000930.]. These could help further refinement of BPD risk estimation of preterm children in order to decrease objective and ethical issues related to implementation of ventilator allocation protocols.
These prediction tools are important even for high-income countries with performant medical systems and relative low incidence of BPD [El Faleh I, Faouzi M, Adams M, Gerull R, Chnayna J, Giannoni E, Roth-Kleiner M; Swiss Neonatal Network. Bronchopulmonary dysplasia: a predictive scoring system for very low birth weight infants. A diagnostic accuracy study with prospective data collection. Eur J Pediatr. 2021 Aug;180(8):2453-2461. doi: 10.1007/s00431-021-04045-8. Epub 2021 Apr 6. Erratum in: Eur J Pediatr. 2021 May 13;: PMID: 33822247; PMCID: PMC8285318.]
Such scores could help in communication process with parents when a shared-decision making process has to be implemented in special cases.
Another important aspect that could be mentioned in decision-making process of resources allocation is ethics. And for extremely premature children especially in the "pre-viable" gestational age group have been published some recent data [Usuda H, Carter S, Takahashi T, Newnham JP, Fee EL, Jobe AH, Kemp MW. Perinatal care for the extremely preterm infant. Semin Fetal Neonatal Med. 2022 Apr;27(2):101334. doi: 10.1016/j.siny.2022.101334. Epub 2022 Apr 15. PMID: 35577715.]
Probably the near future will bring new and more performant algorithms generated by AI [artificial intelligence] and modelling of health and disease trajectories will be more easy to implement.
In severe diseases like malignancies (example used in your paper) there are already AI generated tools to optimize approach of diagnosis, treatment and outcome [Pei Q, Luo Y, Chen Y, Li J, Xie D, Ye T. Artificial intelligence in clinical applications for lung cancer: diagnosis, treatment and prognosis. Clin Chem Lab Med. 2022 Jun 30;60(12):1974-1983. doi: 10.1515/cclm-2022-0291. PMID: 35771735.]
Finally a minor suggestion about unification of citation style in references section. In terms of fonts, links and author listed - in Stoll paper from JAMA you are not listing all authors
Stoll BJ, Hansen NI, Bell EF, Walsh MC, Carlo WA, Shankaran S, Laptook AR, Sánchez PJ, Van Meurs KP, Wyckoff M, Das A, Hale EC, Ball MB, Newman NS, Schibler K, Poindexter BB, Kennedy KA, Cotten CM, Watterberg KL, D'Angio CT, DeMauro SB, Truog WE, Devaskar U, Higgins RD; Eunice Kennedy Shriver National Institute of Child Health and Human Development Neonatal Research Network. Trends in Care Practices, Morbidity, and Mortality of Extremely Preterm Neonates, 1993-2012. JAMA. 2015 Sep 8;314(10):1039-51. doi: 10.1001/jama.2015.10244. PMID: 26348753; PMCID: PMC4787615.
Author Response
- In page 3 you are stating "As survival of preterm infants has increased over the last three decades, there has been a notable increase in the rates of bronchopulmonary dysplasia (BPD) in babies born 22-29 weeks’ (w) gestational age from 32% in 1993 to 40% by 2008 [9]". There are more recent data documenting an increase of BPD likelihood, in very small premature babies, in spite of new, more gentle ventilation modes and tailored approach.
- Thank you for including the recent studies. We specifically included this older study [Stoll, BJ, 2015] to demonstrate that BPD is important to include in a triage protocol due to its increasing incidence in the NICU population. We chose the paper since it gives a comparison of the incidence from 1993 to 2008. The [Jensen EA, 2021] paper you included looks at 24,896 infants up to 29 weeks gestational age and reports that 45.5% of survivors to 36 weeks PMA were diagnosed with BPD. We will incorporate this recent finding in our paper. We did not include the 50% number which was produced from [Chen X, 2019] since they were based on older studies [Lapcharoensap W, 2015] and [Kobaly K, 2008]. We did not include the [Jung YH, 2019] paper since it included only preterm infants with a gestational age less than 28 weeks which may include all babies with BPD.
2. In page 6 you are stating "Infants who survive the first days of life have been shown to have a progressively greater likelihood of survival with each passing NICU day [12]". There are several attempts to generate predictive tools to anticipate outcomes and several biomarkers were proposed for this cohort of children in order to document risk.
- I agree, there are many BPD predictive models that are being studied. These predictive models need to be validated so that we have a consensus on which tools can be used when creating a triage model for neonates. This would greatly facilitate conversations between parents and providers. However, in the model proposed by our institution it was not necessary to predict BPD. Rather, the extra penalty points would be applied only to babies who already have moderate or severe BPD. We agree with you and wanted our paper to convey the importance of further research into neonatal triage. We want to avoid a setting in which neonates are penalized due to a lack of validated triage tools. Also, not having standardized tools leads to variability in resource allocation amongst different institutions.
3. Finally a minor suggestion about unification of citation style in references section. In terms of fonts, links and author listed - in Stoll paper from JAMA you are not listing all authors
-
Thank you for these corrections; we apologize for the oversights and have corrected this